# Combining Materials Obtained by 3D-Printing and Electrospinning from Commercial Polylactide Filament to Produce Biocompatible Composites

**DOI:** 10.3390/polym13213806

**Published:** 2021-11-03

**Authors:** Pablo Romero-Araya, Victor Pino, Ariel Nenen, Verena Cárdenas, Francisca Pavicic, Pamela Ehrenfeld, Guillaume Serandour, Judit G. Lisoni, Ignacio Moreno-Villoslada, Mario E. Flores

**Affiliations:** 1Laboratorio de Polímeros, Facultad de Ciencias, Instituto de Ciencias Químicas, Universidad Austral de Chile, Valdivia 5090000, Chile; pabloromeroaraya@gmail.com (P.R.-A.); victoralfonsopipa@gmail.com (V.P.); ariel.nenen09@gmail.com (A.N.); verena.cardenas@alumnos.uach.cl (V.C.); imorenovilloslada@uach.cl (I.M.-V.); 2Escuela de Odontología, Facultad de Medicina, Universidad Austral de Chile, Valdivia 5090000, Chile; 3Facultad de Medicina, Instituto de Anatomia, Histologia y Patologia, Universidad Austral de Chile, Valdivia 5090000, Chile; francisca.pavicic@gmail.com (F.P.); ingridehrenfeld@uach.cl (P.E.); 4Centro de Estudios Interdisciplinarios del Sistema Nervioso (CISNe), Universidad Austral de Chile, Valdivia 5090000, Chile; 5LeufüLAB, Facultad de Ciencias de la Ingeniería, Instituto de Diseño y Métodos Industriales, Universidad Austral de Chile, Valdivia 5090000, Chile; gserandour@uach.cl; 6Facultad de Ciencias, Instituto de Ciencias Físicas y Matemáticas, Universidad Austral de Chile, Valdivia 5090000, Chile; judit.lisoni@uach.cl

**Keywords:** biocompatible composites, biomimetic composites, PLA 3D-printing, PLA electrospun fibers

## Abstract

The design of scaffolds to reach similar three-dimensional structures mimicking the natural and fibrous environment of some cells is a challenge for tissue engineering, and 3D-printing and electrospinning highlights from other techniques in the production of scaffolds. The former is a well-known additive manufacturing technique devoted to the production of custom-made structures with mechanical properties similar to tissues and bones found in the human body, but lacks the resolution to produce small and interconnected structures. The latter is a well-studied technique to produce materials possessing a fibrillar structure, having the advantage of producing materials with tuned composition compared with a 3D-print. Taking the advantage that commercial 3D-printers work with polylactide (PLA) based filaments, a biocompatible and biodegradable polymer, in this work we produce PLA-based composites by blending materials obtained by 3D-printing and electrospinning. Porous PLA fibers have been obtained by the electrospinning of recovered PLA from 3D-printer filaments, tuning the mechanical properties by blending PLA with small amounts of polyethylene glycol and hydroxyapatite. A composite has been obtained by blending two layers of 3D-printed pieces with a central mat of PLA fibers. The composite presented a reduced storage modulus as compared with a single 3D-print piece and possessing similar mechanical properties to bone tissues. Furthermore, the biocompatibility of the composites is assessed by a simulated body fluid assay and by culturing composites with 3T3 fibroblasts. We observed that all these composites induce the growing and attaching of fibroblast over the surface of a 3D-printed layer and in the fibrous layer, showing the potential of commercial 3D-printers and filaments to produce scaffolds to be used in bone tissue engineering.

## 1. Introduction

Tissue engineering (TE) is devoted to restoring, maintaining, or improving tissue functionality when damage affects the tissue function. Although tissues can self-recover, the use of engineered materials, such as synthetic scaffolds, can improve this process [1,2]. In TE, the use of scaffolds can improve the function of damaged tissues by supporting cell growth and migration, providing an environment with a porous microstructure and adequate pore size distribution, showing biocompatibility and biodegradation, adequate mechanical properties, and shape stability to resist external deformation in order to maintain the integrity of the designed structure. To produce materials for TE meeting these requirements, polymers, ceramics, and metallic materials have been frequently used [3]. In addition, there are a plethora of techniques to produce materials for tissue engineering, such as freeze-drying [4,5], decellularization [6], and surface modification [7].

To mimic the natural environment found in bone, scaffolds can be made using various inorganic compounds such as ceramics and metals. One of these is hydroxyapatite (HA), a well-studied calcium phosphate derivative possessing bioactive and bioresorbable properties to formulate scaffolds for TE; due to this, the inorganic compound is one of the main components in bone tissues [8]. The molecular structure of HA can be modified to include various bioactive ions and signaling molecules. HA can be obtained from natural sources or is synthesized in the laboratory, generating materials with a similar chemical composition but a different porosity, crystal size, and microstructure [9,10]. HA is a substance that can directly bind with live bone after it has been surgically inserted into the human body in cases of bone defects, enhancing appropriate revascularization, stem cell proliferation, and bone regeneration without producing any local or systemic toxicity. Materials designed with pure HA present a limited mechanical strength for load-bearing applications, a problem that can be overcome by including HA in polymeric materials as an additive [11,12].

In this sense, non-toxic and degradable polymeric scaffolds for TE can be prepared by using PLA, acronym of polylactide of poly(lactic acid), a well-known degradable and biocompatible polymer synthesized from lactic acid or lactide, and both are obtained from renewable resources. PLA and related materials are biocompatible and biodegradable due to their undergoing a degradation processes in a physiological environment, mediated by non-enzymatic hydrolysis of the esters in the polymer backbone, generating as degradation products low molecular weight compounds such as lactic and hydroxy hexanoic acids. For instance, lactic acid is the main product obtained after degradation of PLA, which is consumed in normal metabolic pathways, and is expelled from the body in the form of carbon dioxide and water [13,14].

Fused deposition modeling (FDM) is a 3D-printing method that is broadly used in biomedical applications to produce bone models as well as scaffolds with a defined shape [15,16]. In FDM, a fused plastic filament is forced through an extrusion head, and it is deposited on a flat surface into a specific bidimensional pattern. The extrusion head then moves vertically, and a new layer is extruded, which is adhered to the previous deposited layer. A three-dimensional shape is obtained from the accumulation of layers. FDM 3D-printing is convenient as a scaffold fabrication technique. FDM printers are now available at a low cost, which makes them readily available to scientists and physicians. Common filaments such as PLA allows for the good control of scaffold geometry at the macroscale and provides mechanical properties similar to that of bone and other tissues [17,18]. Additionally, the FDM process does not require solvents during printing. However, FDM lacks the resolution needed to obtain functional and biomimetic scaffolds possessing multiple hierarchies of structures on the macro-, micro-, and nanoscale [19,20].

One approach to produce scaffolds for TE is the generation of materials mimicking the native extracellular matrix (ECM), a three-dimensional fibrous-like structure ranging from 10 to 100 nm, composed of a protein matrix synthesized by resident cells in equilibrium with surrounding cells and growth factors [21]. The microenvironment provided by the ECM creates optimal conditions for cell interaction, differentiation, and function [22]. Electrospinning is a suitable technique that allows the preparation of fibrous materials possessing a non-woven structure, mimicking the structure of ECM found in natural tissues [23,24]. Some studies point at the size of electrospinning fibers as influencing the differentiation of neural stem cells [25]. This technique allows the processing of liquid polymeric solutions or solid bulk polymers, allowing the tuning of the composition of the final material by blending polymers with different additives. Several biocompatible polymers are frequently used in electrospinning, such as polycaprolactone (PCL) and PLA, which are both thermoplastic polymers, among other biocompatible and non-biocompatible polymers. In addition, since the polymers are processed in liquid formulations, organic and inorganic compounds such as HA can be easily included [11].

Both techniques mentioned above can be combined to produce scaffolds possessing multiple hierarchic structures [26,27,28], taking advantage of the isolated materials combined in one composite [16]. As said before, electrospinning allows the design of materials with an ECM-like structure and a tuning chemical composition; however, materials with poor mechanical properties to be implanted in the human body are obtained. Meanwhile, commercial 3D-printers and PLA filaments are inexpensive and easy to be implemented, offering the design of materials with complex structures and adequate mechanical properties to be implanted; however, it is difficult to tune the composition of these filaments. Some studies are found in the literature concerning composite design and characterization by combining PLA 3D-printed pieces and electrospun fibers [29,30]. It is found that the addition of a layer of PLA fibers to a 3D-printed PLA piece enhances the mechanical properties of the composite.

In this work, we present the design of polymeric composites by combining PLA-based electrospun fibers supported onto PLA 3D-printed pieces with different pore geometry, in order to obtain a composite mimicking bone tissue, by possessing a porous structure provided by electrospun fibers and supported in a well-defined 3D-printed frame. First, we will describe two digital scaffold designs possessing ellipsoidal and square pore structure, and subsequently the 3D-printing of these designs using a commercial PLA filament in a low-cost commercial 3D-printer. Different PLA fibers have been prepared using recycled PLA from 3D-printed filaments and dichloromethane as a solvent, including or not polyethylene glycol (PEG) and HA as additives. The obtained fibers have been characterized by a scanning electron microscope (SEM), wettability and tensile-test, and the influence of the composition in the final properties of these fibers is discussed. Two types of composites are obtained by combining electrospinning fibers with 3D-printed pieces. The obtained 3D-printed/electrospun composites were characterized by dynamical mechanical analysis (DMA) and SEM, and biocompatibility was tested using the simulated body fluid assay (SBF), and by culturing composites with 3T3 fibroblast cells. The influence of the composition of fibers and 3D-printed pore geometry is discussed.

## 2. Materials and Methods

### 2.1. Materials

Commercial polylactide filament was used to print the scaffolds (1.75 mm diameter, Shenzhen Creality 3D Technology Co. Ltd., Shenzhen, China). Polyethylene glycol (PEG, 100.000 g/mol), 3-(4,5-dimethylthiazol-2-yl)-2,5-diphenyltetrazolium bromide (MTT), and sodium cacodylate were obtained from Sigma (St. Louis, MO, USA). Dichloromethane (DCM), methanol (MeOH), chloroform (CHCl_3_), ethanol (EtOH), sodium chloride (NaCl), sodium bicarbonate (NaHCO_3_), potassium chloride (KCl), dipotassium hydrogen phosphate (K_2_HPO_4_), magnesium chloride hexahydrate (MgCl_2_ × 6H_2_O), hydrochloric acid (HCl, 37 wt.%), calcium chloride dihydrate (CaCl_2_ x 2H_2_O), sodium sulfate (Na_2_SO_4_), and tris(hydroxymethyl)aminomethane (TRIS) were supplied by Merck (Darmstadt, Germany) and used without further purification. Dulbecco’s modified Eagle medium (DMEM, 1.8 mM Ca^+2^) and fetal bovine serum (FBS) were used to cell culture (Gibco). Deionized water was produced in our laboratory. 3T3 fibroblast cells were obtained from primary mouse epidermal cell cultures. Hydroxyapatite (HA) was synthesized following published procedures (see Appendix A) and characterized by TEM (Appendix A) and SEM-EDS (Appendix A) [31,32].

### 2.2. Equipments

Scaffolds were printed using a fused deposition modelling process (FDM) on a commercial 3D-printer (CR-10 Max, Shenzhen Creality 3D Technology Co. Ltd., Shenzhen, China). Electrospinning was done in an apparatus consisting of a 5 mL luer-lock glass syringe (Hamilton Gastight, Franklin, MA, USA) connected to a PTFE tubing with a PEEK luer-lock connector with a flat-end 22G metallic needle, a syringe pump (NewEra Pump Systems Inc, Farmingdale, NY, USA) to control the feeding rates, a plastic plate covered with a rectangular metallic mesh, a high voltage DC power supply (Genvolt High Voltage Ind. Ltd., Bridnorth, United Kingdom) connected to the metal needle and to the ground metallic mesh. SEM images of fibrous materials were obtained in CrossBeam Scanning Electron Microscope (FIB-SEM Auriga Compact, Zeiss, Aalen, Germany). SEM-EDS images were obtained in a variable pressure SEM (PV-SEM EVO MC10, Zeiss, Aalen, Germany) possessing an X-ray microanalysis detector (Oxford Instruments, Abingdon, United King). Wettability of fibrous materials was measured by the sessile drop method in a contact angle meter (HO-IAD-CAM-01, Holmarc Opto-mechanotronics PVT. LTD., Kochi Kerala, India). Thermal and mechanical properties of 3D-printed scaffolds and composites were measured using a dynamic mechanical analyzer (DMA 8000, Perkin Elmer, Waltham, MA, USA). Tensile tests were carried in the fibrous materials by stress–strain tests using a texture analyzer (CT3-1000, Brookfield, Middleboro, MA, USA). Colorimetric MTT assays were performed using a UV-vis spectrophotometer (Bio-MateTM 3S, Thermo Fisher Scientific Inc, Waltham, MA, USA). Simulated body fluid assay was done in a thermostated orbital shaker (WiseCube WIS-20, Witeg Laborthecnik GmbH, Wertheim, Germany).

### 2.3. Methods

#### 2.3.1. Computer-Assisted Design and 3D-Printing of Scaffolds

Two scaffolds with a square and ellipsoidal pore shape were designed using the Inventor Autodesk software (Adobe), and the digital designs are shown in Figure 1. The dimensions of 3D-designed scaffolds are listed in Table 1.

Obtained CAD designs were sliced in a slicer software (Creality Slicer v-4.2) and printed in a commercial 3D-printer (CR-10 Max, Shenzhen Creality 3D Technology Co. Ltd., China), using a commercial PLA filament (1.75 mm, white, Shenzhen Creality 3D Technology Co. Ltd., China). 3D-printer settings used to print the designed scaffolds are listed in Table 2.

#### 2.3.2. Recovery of PLA from 3D-Printer Filaments

PLA was recovered from a virgin 3D-printer filament by dissolution and precipitation. Briefly, 100 g of crushed 1.75 mm filament were added to a 1 L beaker and dissolved with an excess of chloroform. After complete dissolution, the mixture was poured into an excess of cold methanol, inducing the precipitation of the polymer. The resultant mixture was filtered and dried at 30 °C up to constant mass, and then ground in a coffee grinder. The resulting polymeric bulk powder was stored in closed containers.

#### 2.3.3. Electrospinning of PLA and Mixtures with PEG and HA

Four different recovered PLA solutions were prepared using DCM as solvent. One solution was prepared at 10 wt.% of PLA, by dissolving 800 mg of PLA in 5.0 mL of DCM. Moreover, similar solutions were prepared by adding PEG as an additive (2.5 wt.% relative to PLA mass), and HA (2.5 and 5.0 wt.% relative to PLA mass). Mixtures were prepared in 10 mL glass vials and dissolved by magnetic stirring (50 rpm) for 24 h. The preparation of electrospinning mats was done in a room with controlled temperature (20 °C) and relative humidity (40%). Prepared solutions were loaded in a 5.0 mL glass syringe and injected at a speed of 6 mL/h. An electrical potential of 13.0 kV was applied, and the distance of collector-to-needle was kept at 21 cm. A metallic mesh collector was used to collect the mats. After processing 1000 µL of solution, a fibrous mat is obtained in the collector. To evaporate the residual solvent, the obtained materials were left in a laminar flow cabinet for 24 h.

#### 2.3.4. Composite Fabrication by Mixing 3D-Printed and Electrospinning Materials

Both 3D-printing and electrospinning techniques were combined to generate a composite. For the fabrication of the composites, 3D-printed scaffolds were cut in rectangles of 40 × 8.3 × 1.5 mm^3^. Next, electrospun PLA-based fibers were deposited over the 3D-print pieces by placing them over the electrospinning mesh collector. The same experimental conditions described above were applied to deposit fibers over 3D-printed scaffolds. A sandwich-like composite is obtained after adding a second 3D-printed rectangle over the electrospun 3D-print piece. Both faces were stuck by adding a layer of PLA dissolved in chloroform in the extremes of the composite.

#### 2.3.5. 3T3 Fibroblast Culture and Morphological Characterization

3T3 fibroblast cells were obtained from primary mouse epidermal cell cultures. 3T3 cells were expanded in DMEM, supplemented with 10% fetal bovine serum (FBS, Gibco), 1% penicillin-streptomycin (10^4^ UL/mL of penicillin G sodium and 10^4^ mg/mL streptomycin sulfate, Gibco), and 1% fungizone (250 mg/mL amphotericin B, Gibco). The culture was carried out at 37 °C with 5% CO_2_, until sub-confluence. Morphological characterization of 3T3 fibroblasts was done by immunocytochemistry. 3T3 cells were cultured with DMEM 10% supplemented with FBS in a Lab-Teck chamber culture plate over adherent coverlips (Thermo Fisher Scientific Inc, Waltham, MA, USA) by adding 20,000 cell per well. Cells were maintained at 37 °C in CO_2_ atmosphere during 48 h. After that, culture media was removed from the culture plate, and the remaining cells were washed twice with PBS 1X buffer at room temperature. Washed cells were fixed by incubation in 4% paraformaldehyde solution during 20 min at room temperature and were washed three times with PBS 1X buffer. Fixed cells in coverslips were pretreated with metanol-H_2_O_2_ 0.3% during 10 min to permeabilize the cells and blocked the peroxidase activity. Primary antibodies anti-Vimentin (VIM, Cell Signaling) and alpha smooth muscle actin (α-SMA, Dako) were used a dilution of 1:250 in a PBS-BSA 0.5%—saponin 0.3% buffer and incubated overnight at 25 °C in a humid chamber. After that, cells were washed three times with PBS 1X buffer for 10 min, and visualization of fibroblasts markers was done using the biotin/streptavidin-peroxidase method and visualize peroxidase activity by diaminobenzidine tetrahydrochloride as chromogen. Nuclei were counterstained with hematoxylin. A control assay was done by omission of the first antibodies.

#### 2.3.6. 3T3 Adhesion to Composites

A study of the adhesion of 3T3 cells to the surface of the composites was done by culturing cells in the presence of the composites. Pieces of 1 × 1 cm^2^ were introduced in a 24-well culture plate, then, 3 × 10^5^ 3T3 fibroblasts were seeded per well, which was maintained in 10% DMEM media supplemented with FBS. After a certain time of culturing of fibroblast in the presence of the composites (5, 10, and 15 days), both the culture medium and the composite were removed. To study fibroblast morphology, proliferation, and cell adhesion by electron microscopy, composites were gently washed two times with 1 mL of PBS buffer. Cells were fixed at 4 °C with a 2.5% *v/v* glutaraldehyde solution prepared in 0.1 M cacodylate buffer (Sigma Aldrich, St. Louis, MO, USA) for 2 h, and then washed with 0.1 M cacodylate buffer. Dehydration of samples was done using ethanol:water solutions (30, 60, 70, 90, and 100% ethanol content), starting from low to high concentration. A final washing was done using anhydrous ethanol. Remotion of residual ethanol was done using a CO_2_ critical point dryer (Hitachi HCP-2). Samples were stored in a desiccator before SEM analysis.

#### 2.3.7. MTT Assay

The metabolic reduction of 3-(4,5-dimethylthiazol-2-yl)-2,5-diphenyltetrazolium bromide (MTT, Sigma Aldrich, St. Louis, MO, USA), was used to assess the fibroblast viability cultured in the presence of the composites, following the norms for the biological evaluation of medical products by means of cytotoxicity tests (ISO 10993-5) [33]. To carry out this assay, 3 × 10^5^ 3T3 fibroblasts were seeded per well, and were maintained in 10% DMEM supplemented with FBS. Next, the composites were introduced into the culture wells. After a certain amount of time culturing cells with the composites (5, 10 and 15 days), both the culture medium and the composite were removed from the culture plate. Then, an MTT solution (0.55 mg/mL) was added to the culture plate, leaving it in contact with the cells for 4 h. Subsequently, the formazan was solubilized in 1 mL of a mixture of isopropanol and hydrochloric acid in a ratio of 2:10 vol.%, and then the absorbance at 570 nm was measured using a UV-vis spectrophotometer (Bio-MateTM 3S, Thermo Fisher Scientific Inc., Waltham, MA, USA). All tests were carried out in triplicate.

#### 2.3.8. Simulated Body Fluid Assay (SBF)

The SBF solution was prepared according to Kokubo et al. (see Appendix A for ionic composition). Salts were dissolved in deionized water, adjusting the pH to 7.4 using 1.0 M HCl solution. The solution was filtered with a 0.22 µm cellulose filter and stored at 4 °C before use. Rectangular composites of 40 × 8.3 × 1.5 mm^3^ were immersed in 10 mL of SBF solution in plastic containers and were kept in an orbital shaker (WiseCube) at 36.5 °C and 50 rpm. SBF solution was replaced every 4 days with a freshly prepared one. After 30 days, composites were removed from SBF media, gently rinsed with deionized water twice, and dried in a convection oven at 36 °C for 48 h. Samples were stored in a desiccator before analysis.

#### 2.3.9. Characterizations

PLA electrospun fibers were characterized by tensile tests using a universal testing machine (CT3-1000 Texture Analyzer, Brookfield, Middleboro, MA, USA). Wettability of PLA fibers was characterized using a contact angle meter (Holmarc HO-IAD-CAM-01, Kalamassery, Koshi, India), depositing 10 µL of water on the surface of the material. The drop was analyzed using the Image J software possessing the contact angle plugin (NIH, USA). Thermomechanical analysis of 3D-printed scaffolds and composites was done in a Dynamic Mechanical Analyzer (DMA8000, Perkin Elmer, Waltham, MA, USA), under dual cantilever geometry. Storage and loss modulus, and tan δ were measured in a temperature scan program from 25 to 120 °C, heating at 2 °C/min, applying a static force of 2.0 N, 1 Hz of frequency, and 0.05 mm of tension. Glass transition of PLA-based materials was assessed by tan δ maximum position in the temperature scan program. Data were analyzed by OriginPro software. The surface of the obtained materials was studied by scanning electron microscopy (FIB-SEM Auriga Compact, Zeiss, Aalen, Germany), by sticking a small material sample in a SEM stub using a conductive carbon adhesive tape (Ted Pella, Redding, USA). Samples were covered with a thin layer (1 nm) of gold by sputtering (Leica EM ACE 600) and were analyzed by SEM at 3.00 kV. Samples for SEM-EDS analysis were coated with a 10 nm layer of carbon (Leica EM ACE 600) and were analyzed at 20.0 kV. Changes in molecular weight distribution of PLA composites were studied by gel permeation chromatography (GPC, Jasco, Japan), equipped with a refractive index detector (RI-4030, Jasco) and a divinylbenzene-based column (DVB column, Jordi Labs) enclosed in a column oven at 40 °C (CO-4060, Jasco). Around 100 mg of the composite was dissolved in 10 mL of chloroform for liquid chromatography (Merck), keeping the samples in an orbital shaker at 37 °C for 24 h. After that, samples were filtered with 0.22 µm PVDF filters and diluted with chloroform for liquid chromatography at a 5 mg/mL final concentration. Samples were injected into the GPC running in chloroform as a mobile phase at 1 mL/min. Mass average molecular weight (M_w_), number average molecular weight (M_n_), and polydispersity index (PDI), were calculated using ChromNAV-GPC software (Jasco), applying a molecular weight calibration curve constructed using different narrow polymethylmethacrylate (PMMA) as standard (ReadyCal Kit, Polymer Standard Service GmbH).

## 3. Results and Discussion

### 3.1. CAD Design and 3D-Printing

CAD is a powerful tool for the design of scaffolds with simple and complex geometries. Using CAD, we designed two types of scaffolds based on the pore shape, ellipsoidal and square, as shown in Figure 1. These CAD files were transformed into 3D-printer readable files and were printed in a conventional low-cost 3D-printer. Obtained 3d-printed scaffolds are shown in Figure 2. A piece of dimensions 40 × 40 × 1.5 mm^3^ were obtained. In both 3D-printed pieces, a repetitive pore shape was obtained in the whole piece, which can be used to influence the cellular aggregation and nutrient diffusion during the culture of 3D-printed scaffolds with cells [34,35], which also influence tissue formation [36,37]. In addition, the pore size of the 3D-printed scaffolds could influence cell growing, suggesting to some authors that the optimal range is between 200 and 800 µm [35,37]. In the 3D-printed scaffolds, the pore size ranged between 1000 and 1500 µm (Figure 1).

### 3.2. Electrospinning of Recovered PLA

PLA filament used in the 3D-printing of scaffolds was also used to prepare fibrous mats by electrospinning. Previously, a step of recovering of PLA from virgin 3D-print filaments was done by dissolution and precipitation, a simple process for recovering bulk polymers from preformed objects, taking the advantage that PLA is a polymer soluble in organic solvents such as chloroform and dichloromethane, and insoluble in polar solvents such as methanol. This methodology presents some advantages, such as the preservation of the molecular integrity of the polymer compared with other recycling processes for PLA [38,39]. Moreover, this recovery methodology can be also used to recover PLA from failed 3D-print pieces.

Recovered PLA was used to prepare different solutions in DCM as a solvent containing PLA, PEG and HA, and was processed by electrospinning, applying the conditions listed in Section 2.3. Processing pristine PLA solutions prepared at 10 wt.% forms a fibrous material (PLA10); however, a weak material is obtained after detaching from the collector. The addition of PEG (2.5 wt.% related to PLA content) to these solutions improves the formation of a well-formed material over the collector (PLA/PEG), which can be easily handled and detached, as compared with bare PLA fibers. The same trend is observed with solutions prepared with HA at different concentrations (2.5 and 5.0 wt.%), a well-formed material over the collector (PLA/PEG/2.5HA and PLA/PEG/5.0HA). The addition of PEG and HA improves the formation of fibrous PLA mats, even when the PLA concentration is low.

Figure 3 shows the SEM images of PLA electrospun mats. All the obtained materials are composed of non-woven and stacked layers of fibers with diameters around 1 to 5 µm, possessing a similar fibrous and interconnected structure to ECM. However, some differences in the fiber diameter are observed in fibers prepared with PEG and HA. PLA10 material is composed of irregular fibers of a mean diameter of 2.5 µm. Some beads are detected during SEM visualization, which is also consistent with the low handling properties observed during electrospinning of PLA 10 wt.% solution. In PLA/PEG material, a small decrease in the diameter of fibers to 2.0 µm is observed compared with PLA10 fibers. On the contrary, PLA/PEG/2.5HA materials are composed of fibers with a diameter similar to PLA10 fibers, and some small diameter fibers are observed in PLA/PEG/5.0HA materials. Interestingly, in all the obtained electrospun materials, a porous and wrinkled topology is observed on the surface of these fibers, with slight differences in the pore shape, size, and distribution according to the composition of the material. The formation of pores on the surface of electrospun fibers is postulated as a consequence of the high volatility of the solvent used to prepare the solutions (DCM), inducing the microscopic phase separation of the solutions at the surface of the ejected charged jet that occurred very rapidly during electrospinning [40,41]. Another approach to explain the formation of pores in the surface of the fibers is related to moisture condensation when an electrospinning experiment is done under humid environments [42].

Mechanical properties of obtained PLA electrospun mats were measured by tensile test, and the results are shown in Figure 4. Deformation was done up to 50% of the original length of the materials, using a patch of 4.0 cm long. PLA10 material displays a high deformation under low stress, and thus a low Young modulus is measured (0.4 ± 0.2 MPa). Materials formulated with PEG (PLA/PEG) present a Young modulus of 1.3 ± 0.5 MPa, which is stiffer than materials formulated with bare PLA (PLA10). Materials containing HA (PLA/PEG/2.5HA and PLA/PEG/5.0HA) are also stiffer than those prepared without HA, possessing a Young modulus of 1.2 ± 0.6 and 0.7 ± 0.5 MPa, respectively. A decrease in the Young modulus is observed when the concentration of HA increases to 5.0 wt.%, which is explained on the basis of HA dispersion in the PLA-based fibers. Materials formulated with a low concentration of HA allow an adequate dispersion of HA in the polymeric fibrous matrix; thus, an efficient stress transfer is observed during tensile testing. On the contrary, a high load of HA could produce an aggregation of this component in the polymeric matrix, acting as weak points [43,44].

Both PEG and HA are well-known compounds that are frequently added to polymeric material formulations to produce hydrophilic and wettable materials [24,45]. In Figure 5 is shown the contact angle of a water drop on the surface of the PLA electrospun mats. PLA is a hydrophobic polymer, producing a high contact angle of water measured in PLA10 material (128.8 ± 2.2°). A small decrease of the contact angle (121.3 ± 4.9°) is observed in PLA/PEG as compared with pristine PLA electrospun mats. The trend is also observed in PLA/PEG/HA mats, 118.7 ± 2.1° and 116.2 ± 3.0° for materials prepared with 2.5 and 5.0 wt.% HA, respectively. Although it is well-known that PEG is a hydrophilic polymer, the small decrease in the contact angle of PLA materials formulated with PEG is due to the low concentration of PEG used to prepare the fibrous materials.

### 3.3. Composite Design by Combining 3D-Printed Pieces with Electrospinning PLA-Based Fibers

3D-printing and electrospinning techniques were combined to produce 3D-printed/electrospun composites, consisting of two 3D-printed pieces stacked on a central layer of PLA electrospun mat in a sandwich-like structure (Figure 6). Eight materials were obtained after combining the two pieces of the 3D-printed scaffold possessing square and ellipsoidal pores with the four different PLA electrospun mats. For the composite fabrication, a layer of PLA-based fibers was deposited by electrospinning, placing a 3D-printed PLA piece in the electrospinning collector, and then another 3D-printed PLA piece was stacked on top of them. While the 3D-printed scaffolds afford good mechanical properties to the composite, the PLA electrospun mats allow tuning of the material composition. Indeed, an electrospinning deposition of PLA 10 wt.% solution over the 3D-printed PLA pieces failed, whereas it succeeds using PLA/PEG and PLA/PEG/HA solutions (Figure 6).

Dynamic mechanical analysis (DMA) was used to study the mechanical properties of 3D-printed pieces and composites at different temperatures. This technique performs a low deformation into the material, measuring the response of the material to a sinusoidal deformation. In DMA, the modulus is the parameter used to describe the overall resistance of materials to a deformation. The elastic modulus (or storage modulus, E′) measures the elasticity and the ability of the material to store energy; the viscous modulus (or loss modulus, E″) measures the ability of the material to dissipate energy, and the tan δ (or damping factor) is calculated as the ratio of viscous to the elastic response. High values of tan δ are indicative of a high potential of the material to dissipate energy by segmental motion. The damping factor reflects the energy dissipation potential of materials [46]. In Figure 7 is shown the evolution of E′ and tan δ of PLA 3D-printed pieces in a temperature scan from room temperature to 120 °C. During the thermal cycle applied, three clear zones are observed from both E′ and tan δ, and are related to changes in molecular mobility of PLA chains in the material: glassy state, glass transition, and rubbery state, respectively. In the glassy state, the PLA molecules are forming a rigid structure, and only secondary relaxations can occur (γ and β), thus a high E′ value (>10^9^ Pa) is observed in this zone, reflecting the glassy state of PLA at this temperature range. An increase in temperature produces an increase in polymer molecular mobility (α relaxations), and is marked by a considerable drop in the E′ observed in a narrow range of temperatures, also appearing at well-defined tan δ peaks in this temperature range. In addition, tan δ is frequently used to measure the glass transition (T_g_) of polymers in DMA experiments [47,48]. In our assays, the T_g_ of PLA used in 3D-printing is 67–68 °C, similar to reported values applying the same measurement conditions [49,50]. In the rubbery state, all PLA macromolecules have enough mobility to form crystalline domains, then a small increase in the E′ is observed over 90 °C, related to the cold-crystallization of PLA [51,52].

The E′ at room temperature (25 °C) was measured from DMA for the composites and were compared between single, double composite and 3D-print/electrospun composite (Figure 8). Single ellipsoidal 3D-printed pieces present a E′ of 3400 ± 1200 MPa, meanwhile, the single pieces with a square pore shape present a E′ of 4100 ± 2000 MPa. The coupling of two 3D-printed pieces (composite) produces a material with low E′, 1400 ± 490 MPa for an ellipsoidal composite, and 730 ± 40 MPa for square composite, when compared with single pieces. Small changes are observed with tan δ (ellipsoidal single 2.1 ± 0.1; square single 1.8 ± 0.1; ellipsoidal composite 2.2 ± 0.1; square composite 1.9 ± 0.1). Similar results in E′ have been observed in 3D-printed PLA blends with a different thickness [53]. Interestingly, the obtained E′ for 3D-printed PLA pieces, >10^3^ MPa, are similar to values obtained by the DMA characterization of the viscoelastic behavior of human cortical bone [18]. E′ also was measured for 3D-printed/electrospun composites, and the obtained values are low in comparison with single 3D-printed pieces. However, an increase in E′ is observed in square composites containing PLA/PEG/5.0HA, when this sample is compared with composite containing PLA10 fibers. Fibers used to produce the 3D-printed/electrospun composite reinforce the mechanical properties of the composites; thus, tuning the mechanical properties of composites is done by tuning the composition of electrospun fibers.

### 3.4. PLA-Based Composites as Scaffolds

#### 3.4.1. SFB Assay

The potential of the 3D-printed/electrospun composites to induce apatite formation in their surface when immersed in an acellular ionic solution and the simulated body fluid assay (SBF) were used as an in-vitro bioactivity test to study the bone-bonding ability of designed materials [54]. For the obtained composites, we studied the mechanical properties, molecular weight, and calcium derivative depositions after 30 days of immersion of the composites in SBF media. DMA was used to measure the E′ of the composites at body temperature, 37 °C, being PLA in their glassy state, and compared before (t_0_) and after immersion in SBF media for 30 days (t_30_) (Figure 9). A control sample was fabricated using two 3D-printed pieces bonded in the absence of a layer of electrospun fibers. The control samples having an ellipsoidal and square pore shape present a E′ at 37 °C of 1610 ± 315 MPa and 707 ± 37 MPa, respectively, as can be seen in the Figure 9. The E′ of 3D-printed/electrospun composites did not vary significantly as compared with control samples. In t_30_ samples is observed an increase in E′ compared with samples at t_0_, which is more pronounced in composites possessing square pores, and is less pronounced in the ellipsoidal. The increase in E′ is related to the formation of a layer of calcium derivatives on the surface of the materials [12,55].

Molecular weight of 3D-printed/electrospun composites immersed in SBF at t_0_ and t_30_ was studied by GPC, estimating the molecular weight by using narrow PMMA standards. Considering that the mass of the composite is mainly the 3D-printed structure, the results shown in Figure 10 are related to the molecular weight of the PLA used in the fabrication of 3D-printed structures. 3D-printed/electrospun composites at t_0_ present a M_w_ of 1.0–1.6 × 10^5^ Da, with PDI ranging from 1.69 to 1.75. No significant differences in M_w_ are observed at t_30_; thus, no evidence of PLA degradation after 30 days was observed.

The deposition of calcium derivatives over the composites during SBF assay was studied by SEM-EDS (see Appendix A). The analysis was done qualitatively in some punctual zones due to the composites here studied did not present a flat surface, required parameter for quantitative SEM-EDS analysis [56]. EDS measurements in 3D-printed/electrospun composites formulated with PLA/PEG/2.5HA and PLA/PEG/5.0HA revealed the presence of Ca and P, with ratios exceeding the expected value for HA, i.e., 5.9–8.9 vs. 2.2 (wt.% ratio), respectively. This might indicate of the formation of calcium derivatives over the surface of composites as expected from the SBF assay. The Ca exceeding the amount might be related to the presence of calcium salts used in the SBF assay.

#### 3.4.2. 3T3 Cell Culture in the Presence of Composites

To assess the biocompatibility of the obtained PLA-based composites, we cultured the obtained 3D-printed/electrospun composites in the presence of 3T3 fibroblast cells, a broadly used in-vitro assay for biomaterials [57,58]. We selected fibroblasts to assess the in-vitro biocompatibility of the composites due to these being the most common cells found in the connective tissues of mammalians [59]. Fibroblasts are also important because they can synthesize specific macromolecules to build an extracellular matrix, thus producing the structural framework for tissues and organs [60]. The 3T3 fibroblasts used in this study were characterized by an inverted microscopy and immunocytochemistry (ICC), and the results are shown in Figure 11. Inverted microscopy images (Figure 11A) show the characteristic elongated and interconnected shape of 3T3 fibroblasts when they are forming a monolayer of cells attached to the culture plate. Hematoxylin staining (Figure 11B) allow it to mark the nuclei of 3T3 cells, showing the characteristic ellipsoidal shape of fibroblast nuclei enclosing two or more nucleolus. By using specific antibodies for fibroblast cell lineages, we identified structural proteins in the cytoskeleton, such as actin (α-SMA, Figure 11C) and vimentin (VIM, Figure 11D), which are proteins associated with the mesodermal origin of the 3T3 fibroblast [61].

An MTT assay was done to assess the viability of 3T3 fibroblasts remaining after culturing these cells in the presence of 3D-printed/electrospun composites at three different times (5, 10 and 15 days), and the results are shown in Figure 12 as percentual cell viability. The results check that the non-toxic nature of designed composites, due to cell proliferation, is observed during culturing of 3T3 cells with composites, obtaining disperse average values, and no significant differences in composites at different times of culture were observed. However, when cell viability of the 3D-printed/electrospun composites is compared with a control composite, a boost in the growth of cells cultured in the presence of ellipsoidal composites is observed, thus promoting proliferation of 3T3 cells during culturing. On the contrary, a low cell viability is observed during the culture of composites, which possesses a square pore shape.

The attachment and morphology of 3T3 fibroblast cultured in the designed 3D-printed/electrospun composites were studied by SEM. For brevity, some selected images are presented in Figure 13 in order to show that fibroblasts adhered in the 3D-printed piece and in the electrospun fibers of the composite. Although we used a commercial 3D-printer filament to print the scaffolds, in all the composites cultured with 3T3 cells we observed cells attached to the main 3D-printed structure (Figure 13A,B), showing the potential of commercial PLA to produce biocompatible materials [62]. The morphology of cells found spread over the surface of 3D-printed pieces of the composite are a blend between some apoptotic cells showing a round shape and a shrunk nucleus, besides a well-formed and interconnected network of fibroblasts linked by polarized filipodia protrusions [63,64,65]. Some cells are found in the fibrous layer of the composites after 5 days of culture (Figure 13C,D). Cells found in the ellipsoidal 3D-printed/electrospun composite prepared with a PLA10 fibrous layer (Figure 13C) are in a small zone of the fibrous layer; however, the cells found in the square composite prepared with PLA/PEG/5.0HA fibers are found to be well spread over the scanned area (Figure 13D), showing the potential of HA to induce cell attaching and spreading over the surface of the fibers.

## 4. Conclusions

Both electrospinning and 3D-printing technologies can be mixed to obtain composites meeting requirements for tissue engineering. By computer-assisted design it was possible to obtain scaffolds with a controlled pore shape that are easily printable in commercial 3D-printers working with PLA-based filaments and a well-known biocompatible and biodegradable polymer. The 3D-printer filament was also used to obtain the polymer used to prepare porous fibers by electrospinning, and the mechanical, morphological, and wettability properties of these fibers is improved by adding small amounts of PEG and HA as additives. A composite was obtained after mixing two layers of 3D-printed scaffolds with a central layer of PLA-based fibers. These composites showed a reduced storage modulus as compared with bare 3D-printed pieces; thus, a soft composite is obtained after blending 3D-printed pieces with electrospun fibers. The culture of these composites with 3T3 cells showed a high cell viability in those prepared with 3D-printed/electrospun composites possessing an ellipsoidal pore shape, and cells were able to attach and grow both over the surface of the 3D-printed layer and electrospun fibers. The results presented in this work shows the potential of commercial PLA 3D-printer filaments to obtain biocompatible and biomimetic composites to be used in bone tissue engineering.

## Figures and Tables

**Figure 1 polymers-13-03806-f001:**
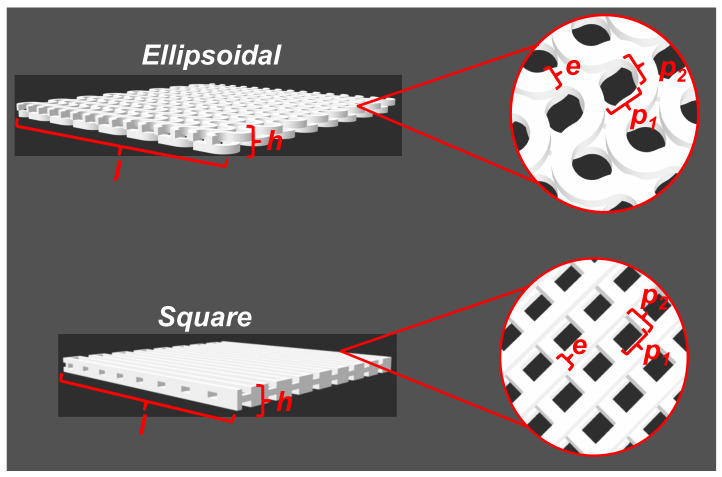
Scaffolds obtained by CAD. Values related with scaffold dimensions are listed in Table 1.

**Figure 2 polymers-13-03806-f002:**
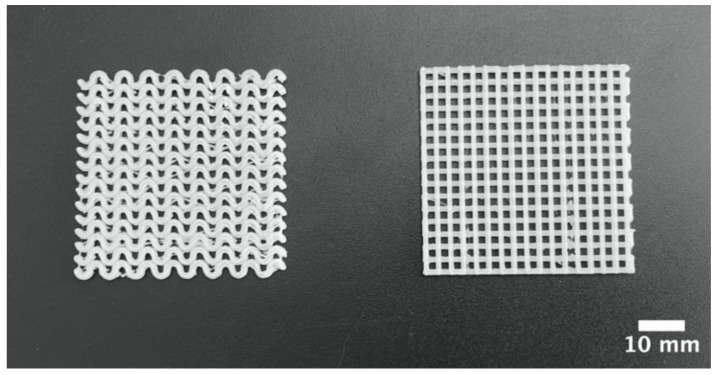
Macrostructure of ellipsoidal and square pore shape PLA pieces obtained through 3D-printing.

**Figure 3 polymers-13-03806-f003:**
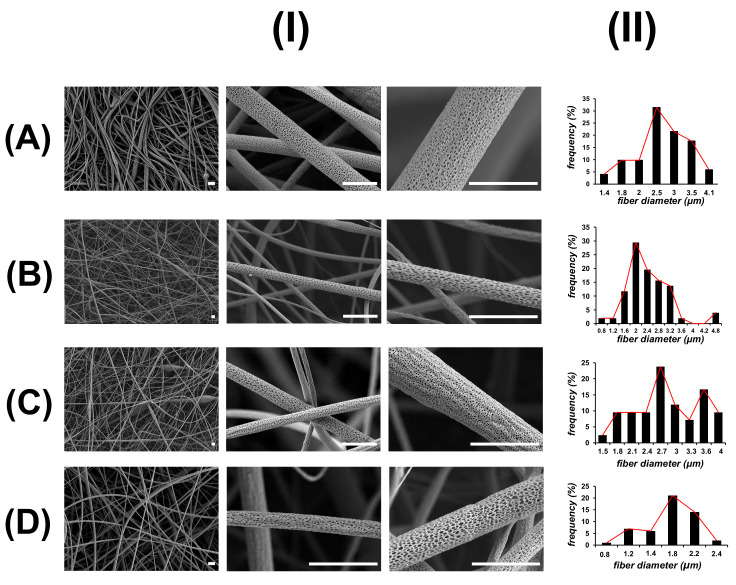
SEM images (**I**) and size distribution of fiber diameter (**II**) of PLA-based electrospun fibers: (**A**) PLA10, (**B**) PLA/PEG, (**C**) PLA/PEG/2.5HA, and (**D**) PLA/PEG/5.0HA. Scale bar is 5 µm.

**Figure 4 polymers-13-03806-f004:**
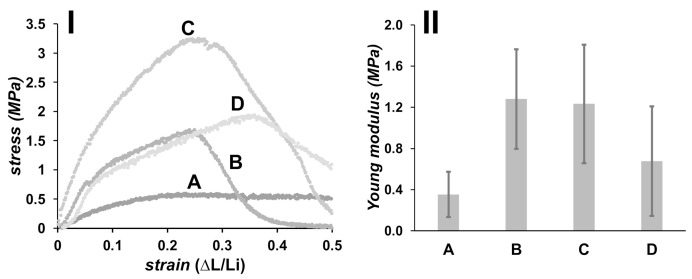
Tensile stress–strain plot (**I**), and Young modulus for PLA-based fibers (**II**): (A) PLA10, (B) PLA/PEG, (C) PLA/PEG/2.5HA, and (D) PLA/PEG/5.0HA.

**Figure 5 polymers-13-03806-f005:**
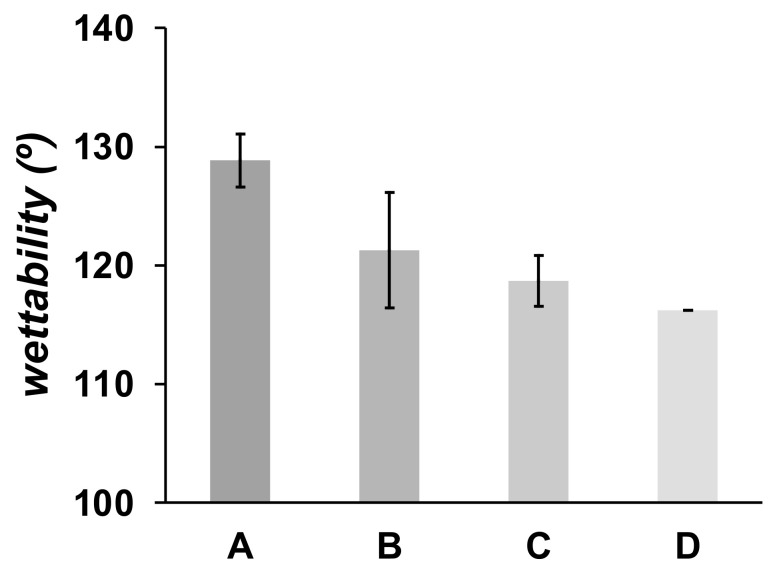
Wettability of electrospun fibers. (A) PLA10, (B) PLA/PEG, (C) PLA/PEG/2.5HA, and (D) PLA/PEG/5.0HA.

**Figure 6 polymers-13-03806-f006:**
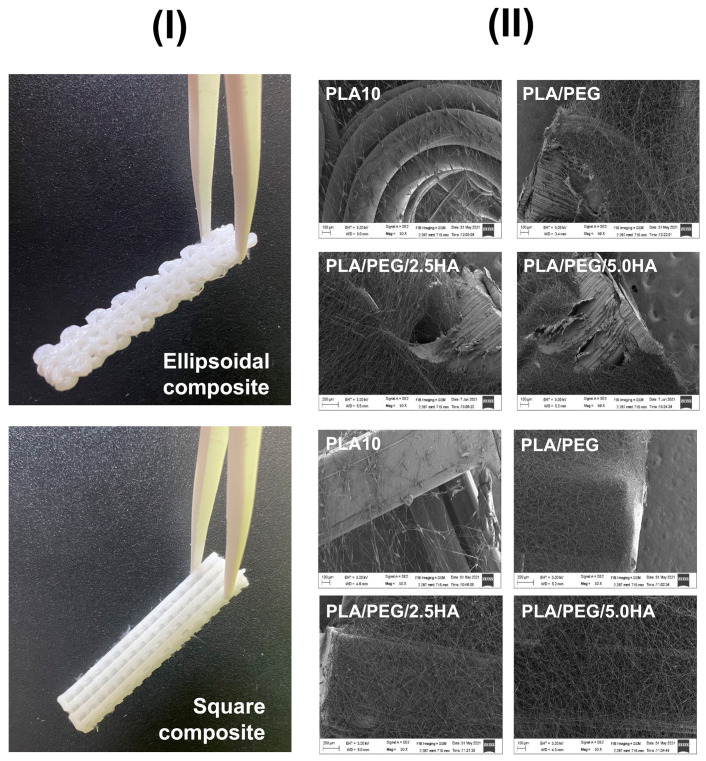
(**I**) Images of 3D-printed/electrospun composites obtained after combining 3D-printed PLA pieces with ellipsoidal and square pores with PLA-based fibers. (**II**) SEM images of the fibrous layer of PLA-based electrospun composites deposited over the surface of 3D-printed pieces.

**Figure 7 polymers-13-03806-f007:**
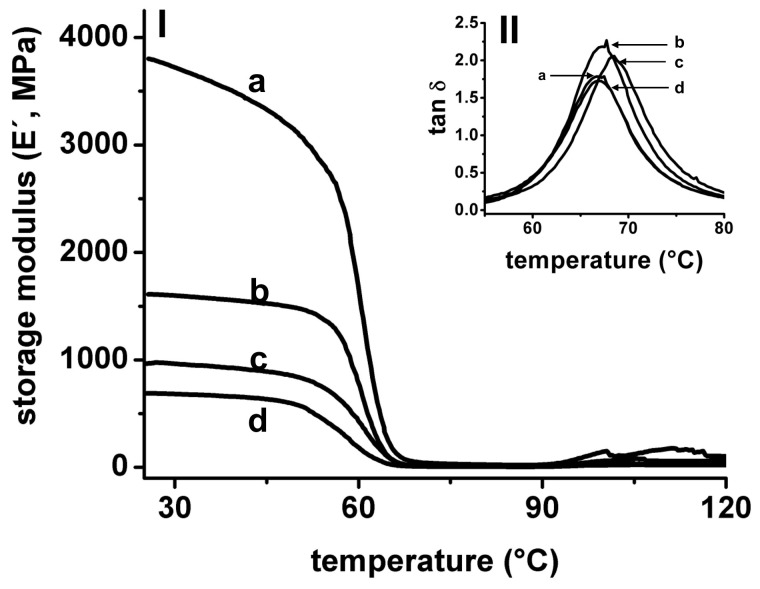
Storage modulus (E′) (**I**) and tan δ (**II**) measured in DMA from a temperature scan for single square (a), and single ellipsoidal (b) 3D-printed pieces. The E′ and tan δ of two 3D-printed pieces with ellipsoidal (c) and square (d) pores bonded forming a double composite are also shown.

**Figure 8 polymers-13-03806-f008:**
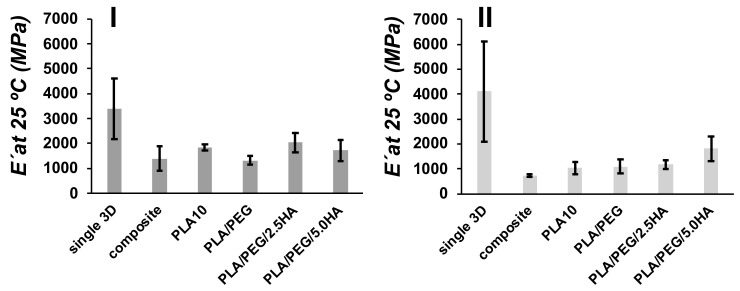
Comparison of E′ for ellipsoidal (**I**) and square (**II**) pore shape 3D-printed materials at 25 °C for single 3D-printed pieces (single 3D), two 3D-printed pieces bonded (composite), and 3D-printed/electrospun composites (PLA10, PLA/PEG/, PLA/PEG/2.5HA, and PLA/PEG/5.0HA).

**Figure 9 polymers-13-03806-f009:**
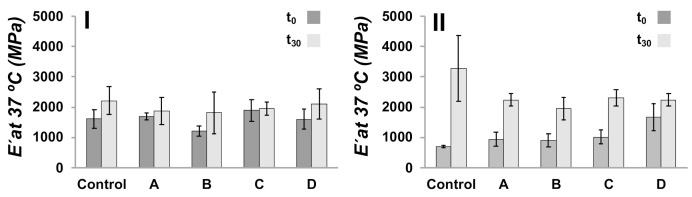
E′ at 37 °C of ellipsoidal (**I**) and square (**II**) 3D-printed/electrospun composites immersed in SBF media for 30 days. A control sample is a composite prepared without covering the surface of 3D-printed pieces with fibers, and A-D are those composites prepared with a layer of PLA-based fibers: (A) PLA10, (B) PLA/PEG, (C) PLA/PEG/2.5HA, and (D) PLA/PEG/5.0HA.

**Figure 10 polymers-13-03806-f010:**
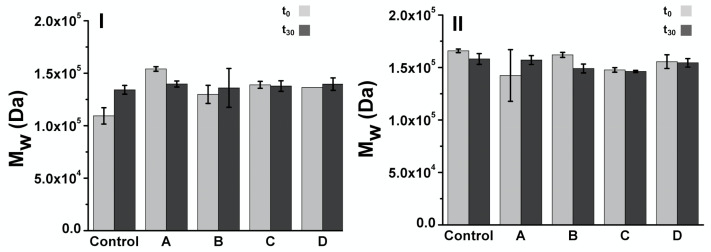
Molecular weight estimated by GPC for of 3D-printed/electrospun ellipsoidal (**I**) and (**II**) square composites immersed in SBF media for 30 days A control sample is a composite prepared without cover the surface of 3D-printed pieces with fibers, and A-D are those prepared with PLA based fibers: (A) PLA10, (B) PLA/PEG, (C) PLA/PEG/2.5HA, and (D) PLA/PEG/5.0HA.

**Figure 11 polymers-13-03806-f011:**
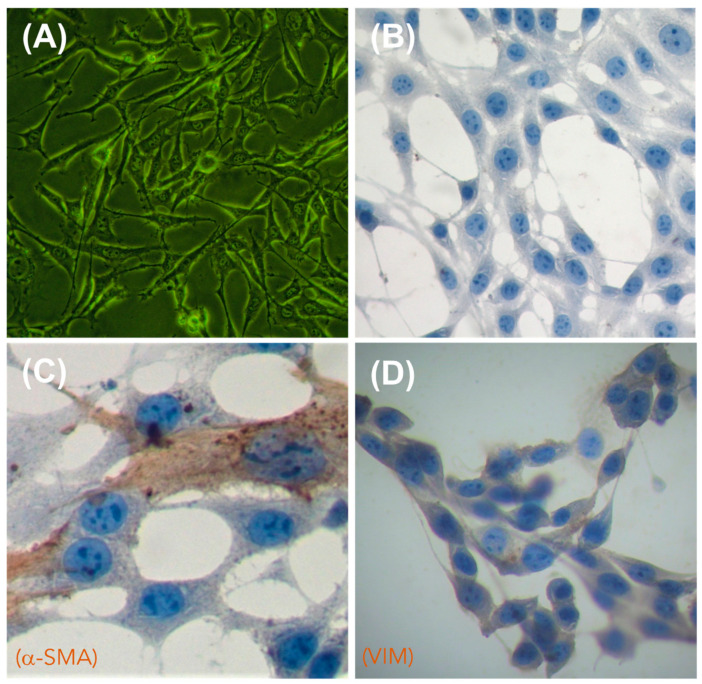
Morphological characterization of 3T3 fibroblasts by inverted microscopy (**A**) and immunocytochemistry (ICC) (**B**–**D**). (**A**) Cells observed by inverted microscopy; (**B**) negative ICC control (omission of antibody); (**C**) visualization of alpha smooth muscle actin (α-SMA, Dako, dilution 1:250), used to mark the cytoskeleton of 3T3 fibroblast; (**D**) visualization of vimentin (VIM, Cell signaling, dilution 1:250), used to mark the cytoskeleton of 3T3 fibroblast. Cells were counterstained with hematoxylin. Images were captured at 40× magnification.

**Figure 12 polymers-13-03806-f012:**
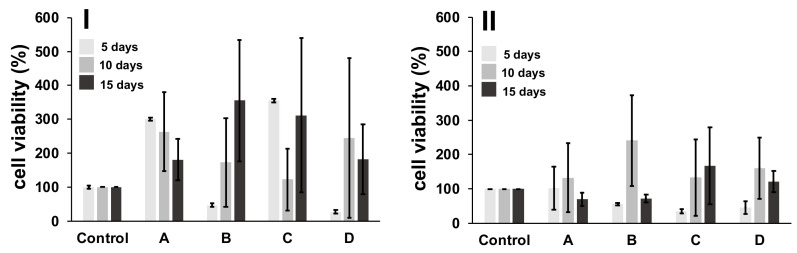
MTT assay for cells after contact with 3D-printed/electrospun ellipsoidal (**I**) and square (**II**) composites. A control sample is a composite prepared without cover over the surface of 3D-printed pieces with fibers; and A-D are those prepared with PLA-based fibers: (A) PLA10, (B) PLA/PEG, (C) PLA/PEG/2.5HA, and (D) PLA/PEG/5.0HA.

**Figure 13 polymers-13-03806-f013:**
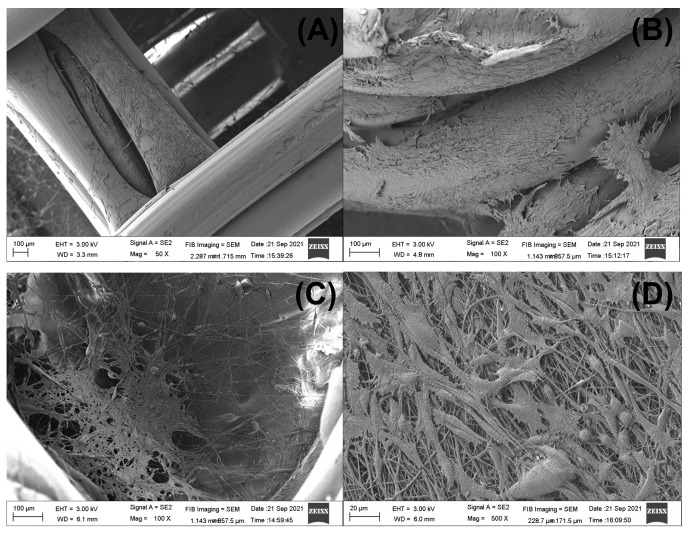
Selected SEM images of 3D-printed/electrospun composites cultured for 5 days with 3T3 cells. Fibroblast adhered to the 3D-printed piece are showed in (**A**,**B**). ((**A**) is a square 3D-printed/electrospun composite with PLA10 fibers, and (**B**) is the ellipsoidal composite with PLA/PEG/5.0HA fibers). The fibroblasts which adhered to the fibrous layer of the composite are showed in (**C**,**D**). ((**C**) is an ellipsoidal composite with PLA10 fibers, and (**D**) is a square composite with PLA/PEG/5.0HA fibers).

**Table 1 polymers-13-03806-t001:** CAD scaffolds dimensions.

Dimensions^1^	Ellipsoidal (mm)	Square (mm)
Length (*l*)	40 mm	40 mm
Pore size (*p*)	*p1*: 1.5 mm	*p1:* 1.5 mm
	*p2*: 1.3 mm	*p2:* 1.5 mm
Height (*h*)	1.5 mm	1.5 mm
Filament thickness (*e*)	1.0 mm	1.0 mm

**Table 2 polymers-13-03806-t002:** 3D-printer parameters used to print.

Parameters	Value
Nozzle diameter	0.4 mm
Layer height	0.1 mm
Initial layer height	0.1 mm
Line width	0.1 mm
Print speed	100 mm/s
Printing temperature	200 °C
Print plate temperature	60 °C
Infill pattern	Lines/Concentric
Type of adhesion of the printing plate	Raft

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
