# Peer review of "Combining Materials Obtained by 3D-Printing and Electrospinning from Commercial Polylactide Filament to Produce Biocompatible Composites"

_polymers, 2021, doi:10.3390/polym13213806_

Round 1

Reviewer 1 Report

The current manuscript provides an interesting account of two fabrication technique combined together to achieve a hybrid platform. The work is well conducted and the results are well reported. I recommend following revision to the manuscript:

The concern that I have with the final results is that the authors stated "A sandwiched composite has been obtained by blending two layers of
3D-printed composite with a central mat of PLA fibers. The composite presented a reduced storage modulus as compared with a single 3D-print piece." Authors should comment on the performance of the composite scaffold vs that of two 3D printed layers combined or even printing a single 3D printed structures of the same thickness or even introducing a 3D printed layer of same thickness as the electrospun fibers - there must be a strong explanation or motivation for combining the two systems together and should be backed by evidence from controls.

Author Response

Reviewer 1:

The current manuscript provides an interesting account of two fabrication technique combined together to achieve a hybrid platform. The work is well conducted and the results are well reported.

  • I recommend following revision to the manuscript: The concern that I have with the final results is that the authors stated "A sandwiched composite has been obtained by blending two layers of 3D-printed composite with a central mat of PLA fibers. The composite presented a reduced storage modulus as compared with a single 3D-print " Authors should comment on the performance of the composite scaffold vs that of two 3D printed layers combined or even printing a single 3D printed structures of the same thickness or even introducing a 3D printed layer of same thickness as the electrospun fibers

Response: In the manuscript is characterized the mechanical behavior of the electrospun mats by tensile test, and the performance of simple and double 3D-printed pieces and 3D-printed/electrospun composites measured by dynamical mechanical analysis (DMA). This technique is useful to assess the stiffness of a plastic material by measuring the storage modulus (E´). In the sent manuscript we discussed the mechanical performance of single and double 3D-printed pieces (Figure 8), and we compare E´ with 3D-printed/electrospun composites. We highlight in the manuscript that composite present a reduced storage modulus as compared with single and double 3D-printed pieces, and the importance of improving the stiffness of PLA-based materials.

We added in the 3.3 section a sentence related with the mechanical performance of the PLA-based materials described in the manuscript:

“Fibers used to produce the 3D-printed/electrospun composite reinforce the mechanical properties of the composites, thus tuning the mechanical properties of composites is done by tuning the composition of electrospun fibers.”

  • There must be a strong explanation or motivation for combining the two systems together and should be backed by evidence from controls.

Response: In the final paragraph of the introduction section, we added a new sentence pointing our motivation to combine the materials obtained from electrospinning and 3D-printing in one composite:

"In this work, we present the design of polymeric composites by combining PLA-based electrospun fibers supported onto PLA 3D-printed pieces with different pore geometry, in order to obtain a composite mimicking bone tissue, by possessing a porous structure provided by electrospun fibers and supported in a well-defined 3D-printed frame”

Reviewer 2 Report

The authors present an interesting manuscript with potential, although the novelty in the field is limited. Authors must implement major remodeling of the manuscript for the manuscript to be published successfully:

-Firstly, the authors present a general title that does not have the novelty that the manuscript is supposed to have.

-The summary is absolutely inadequate. Authors have to put data, not just say things that are already known. Those should not digress in the summary.

-The keywords are few and without news.

-The production is too long and with too much rambling. The authors must focus on the state of the art and justify the spirit of the study. The references used must be current, this field of study changes a lot and there are countless studies.

-The introduction must have a translational character, these studies are justified by the translation and the extensibility in preclinical models.

-Material and methods are limited, please include more details from in vivo studies.

-The authors must justify the sample size, mentioning why they can reach these conclusions with such a limited sample size.

-The authors show adequate results, but the figure legends are non-existent.

-A delicate point is the study of cell viability. Authors must show images of cell cultures. MTT studies are appropriate and should be included. -The authors mention the activity of the fibroblast. Authors must show images, and use immunohistochemical techniques to phenotype them.

-The authors must include an appropriate and specific discussion section, where they justify the novelty and applicability.

-The conclusions must be based on the results and not just intentions.

-The authors must improve their use of English grammar very extensively.

-Check the references.

-The authors should include a small systematic review of the main models that allow them to contribute the novelty of the manuscript. 

Author Response

Reviewer 2:

The authors present an interesting manuscript with potential, although the novelty in the field is limited. Authors must implement major remodeling of the manuscript for the manuscript to be published successfully:

  • Firstly, the authors present a general title that does not have the novelty that the manuscript is supposed to have.The summary is absolutely inadequate. Authors have to put data, not just say things that are already known. Those should not digress in the

Response: In order to avoid confusions, we changed the title of the manuscript:

“Combining materials obtained by 3D-printing and electrospinning from commercial PLA to produce biocompatible composites”

The summary/abstract also has been modified to clarify the intention of our work:

“We observed that all these scaffolds induce the growing and attaching of 3T3 fibroblast over the surface of 3D-printed layer, and also in the fibrous layer, showing the potential of commercial 3D-printers and filaments to produce scaffolds to be use in bone tissue engineering”

  • The keywords are few and without news.

Response: We changed keywords to adequate ones, as reviewer suggest.

Keywords: Biocompatible composites; biomimetic composites; PLA 3D-printing; PLA electrospun fibers”

  • The production is too long and with too much rambling. The authors must focus on the state of the art and justify the spirit of the study. The references used must be current, this field of study changes a lot and there are countless studies.
  • The introduction must have a translational character, these studies are justified by the translation and the extensibility in preclinical models.
  • Material and methods are limited, please include more details from in vivo
  • The authors must justify the sample size, mentioning why they can reach these conclusions with such a limited sample size.

Response to comments 3, 4, 5 and 6: The response of these four points commented by the reviewer is detailed below.

We consider that Reviewer 2 is confusing the term in-vivo with in-vitro. The former is an assay using animals, and the latter an assay with well-known cells such as fibroblast, or by using solutions mimicking the environment found in fluids. Culturing with cells is a well-known assay to assess biocompatibility of polymeric materials. Hence, we do not perform biocompatibility assays with animals, then we cannot reply to the comments in points 4, 5, and 6. Thus, the comment related with changing the introduction to a translational character cannot be done due to we present in this work the designing of a composite to be use in a future in bone tissue engineering.

To avoid confusions, we modified the final paragraph of the introduction section:

“The obtained 3D-printed/electrospun composites were characterized by dynamical mechanical analysis (DMA) and SEM, and biocompatibility was tested using the simulated body fluid assay (SBF), and by culturing composites with 3T3 fibroblast cells. The influence of the composition of fibers and 3D-printed pore geometry is discussed”

  • The authors show adequate results, but the figure legends are nonexistent.

Response: All the figures in the sent manuscript present an adequate description.

  • A delicate point is the study of cell viability. Authors must show images of cell cultures. MTT studies are appropriate and should be included.

Response: In the submitted manuscript we included MTT studies of 3T3 cells used (Figure 11 in first manuscript version). Methodology to culture these cells is described in the methods section (2.2), and MTT results are shown in the Figure 12 in the corrected version.

  • The authors mention the activity of the fibroblast. Authors must show images, and use immunohistochemical techniques to phenotype them.

Response: We included in the manuscript a new Figure (Figure 11) showing the morphological characterization of 3T3 fibroblasts cells used in this work by inverted microscopy and immunocytochemistry, together with an explanation of the choice of fibroblast as an in-vitro assay for biocompatibility:

“To assess the biocompatibility of the obtained PLA-based composites, we cultured the obtained 3D-printed/electrospun composites in the presence of 3T3 fibroblast cells, a broad used in-vitro assay for biomaterials [57,58]. We selected fibroblasts to assess the in-vitro biocompatibility of the composites due to these cells are the most common cells found in connective tissues of mammalians [59]. Fibroblast also are important due to their can synthesize specific macromolecules to build an extracellular matrix, thus producing the structural framework for tissues and organs [60]. The 3T3 fibroblast used in this study were characterized by inverted microscopy and immunocytochemistry (ICC), and results are shown in Figure 11. Inverted microscopy images (Figure 11A) show the characteristic elongated and interconnected shape of 3T3 fibroblast when they are forming a monolayer of cells attached to the culture plate. Hematoxylin staining (Figure 11B) allow it to mark the nuclei of 3T3 cells, showing the characteristic ellipsoidal shape of fibroblast nuclei enclosing two or more nucleolus. By using specific antibodies for fibroblast cell lineages, we identified structural proteins in the cytoskeleton such as actin (a-SMA, Figure 11C) and vimentin (VIM, Figure 11D), proteins associated with the mesodermal origin of the 3T3 fibroblast [61]”

  • The authors must include an appropriate and specific discussion section, where they justify the novelty and applicability.

Response: The discussion is included in the main text of the results and discussion section.

  • The conclusions must be based on the results and not just intentions.

Response: We adapted the conclusions with sentences according to the results presented in the manuscript.

  • The authors must improve their use of English grammar very extensively.

Response: English grammar has been corrected, as the reviewer suggest.

  • Check the references.

Response: References has been checked and formatted using the Polymers style.

  • The authors should include a small systematic review of the main models that allow them to contribute the novelty of the manuscript.

Response: The novelty of the manuscript is related with using commercial PLA filaments to produce both 3D-printed and electrospun materials. To highlight this novelty, we changed the title of the article, and we added new sentences in the final paragraph of the introduction section.

Round 2

Reviewer 2 Report

The manuscript has improved. The authors' responses are adequate.